# An Efficient Group-Based Control Signalling within Proxy Mobile IPv6 Protocol

**Safwan M. Ghaleb** [1,*,†] ⓘ **, Shamala Subramaniam** [1,2,†] **, Mukhtar Ghaleb** [3,4,†] **and Ali Mohamed E. Ejmaa** [5,†]

1 Department of Communication Technology and Network, Universiti Putra Malaysia, 43400 UPM Serdang, Selangor D.E., Malaysia
2 Sports Academy, Universiti Putra Malaysia, 43400 UPM Serdang, Selangor D.E, Malaysia
3 College of Arts and Sciences, University of Bisha, Bisha 67714, Al-Namas, Saudi Arabia
4 Faculty of Computer Sciences and Information Technology, Sana'a University, Sana'a 15542, Yemen
5 Biotechnology Research Center, Tripoli 30313, Libya
* Correspondence: gs41921@student.upm.edu.my; Tel.: +60-113-934-1291
† These authors contributed equally to this work.

**Abstract:** Providing a seamless handover in the Internet of Thing (IoT) applications with minimal efforts is a big challenge in mobility management protocols. Several research efforts have been attempted to maintain the connectivity of nodes while performing mobility-related signalling, in order to enhance the system performance. However, these studies still fall short at the presence of short-term continuous movements of mobile nodes within the same network, which is a requirement in several applications. In this paper, we propose an efficient group-based handoff scheme for the Mobile Nodes (MNs) in order to reduce the nodes handover during their roaming. This scheme is named Enhanced Cluster Sensor Proxy Mobile IPv6 (E-CSPMIPv6). E-CSPMIPv6 introduces a fast handover scheme by implementing two mechanisms. In the first mechanism, we cluster mobile nodes that are moving as a group in order to register them at a prior time of their actual handoff. In the second mechanism, we manipulate the mobility-related signalling of the MNs triggering their handover signalling simultaneously. The efficiency of the proposed scheme is validated through extensive simulation experiments and numerical analyses in comparison to the state-of-the-art mobility management protocols under different scenarios and operation conditions. The results demonstrate that the E-CSPMIPv6 scheme significantly improves the overall system performance, by reducing handover delay, signalling cost and end-to-end delay.

**Keywords:** proxy mobile IPv6; IP-WSN; mobility management protocols; handover latency; clustered PMIPv6 protocols; signalling cost

## 1. Introduction

With the emergence of Internet of Thing (IoT) and the Machine-to-Machine (M2M) networks, designing efficient protocols for connecting such networks with the Internet has become more pertinent. In this context, IPv6 and Low Power over Personal Area Network (6LoWPAN) have recently been introduced to fully integrate low-power networks such as Wireless Sensor Network (WSN) with the Internet. However, this integration, coupled with the exponential growth of applications that require mobility support, has generated the need for developing mobility management protocols. Hence, the main aim of mobility management is to maintain the connectivity of mobile node when roaming among different networks. To address such an issue, several mobility management protocols have come into existence with the ultimate goal of providing an efficient and seamless movement of nodes

among networks. In general, the mobility management protocols are categorised into two classes: host-based protocols and network-based protocols. In the host-based protocols [1–5], the MN is required to be involved in the mobility process even when the Network Mobility (NEMO) is supported for the MNs that move in a group by the Mobile Router entity (MR). This involvement leads to increasing the handoff and the MN complexity, which in turn lower the system performance [6].

　To solve this issue, network-based protocols have been proposed [7–12]. In these protocols, the handoff signalling burden is transferred to new entities called Local Mobility Anchor (LMA) and Mobile Access Gateway (MAG) and therefore the Mobile Node (MN) is shielded from the mobility related signalling when it moves between different networks. Although being efficient in terms of power consumption, the network-based protocols do not show the same level of efficiency in terms of handover latency and signalling cost as they usually induce long handover latency and high signalling cost due to the individual processing of the MNs handoff operations. Consequently, group-based technique have been considered by several solutions in order to overcome the issues of signalling cost and handover latency associated with the previous studies, especially when the handover is triggered frequently in a short time for a group of MNs [13–16]. The proposed studies aim at grouping the MNs' control messages to minimise the handoff latency and signalling cost through utilising the similarity in the MNs' movement patterns. Received Signal Strength (RSS) is utilised by most of these studies to discover the MNs' movement similarity during their roaming. According to the above explanation, the main issue that should be considered during the MNs' handoff process is: designing an efficient scheme that has the ability to process the MNs' handoff simultaneously without any service disruption. Thus, the aim of this paper is to propose an efficient scheme for simultaneously moving MNs or MNs having their handoff simultaneously triggered, as shown in Figure 1. The proposed Enhanced Cluster Proxy Mobile IPv6 (E-CPMIPv6) scheme achieves this by utilising the Signal-to-Noise Ratio (SNR) of MNs in order to group the MNs' mobility-related signalling and to discover the MNs that will perform the handoff at the same time.

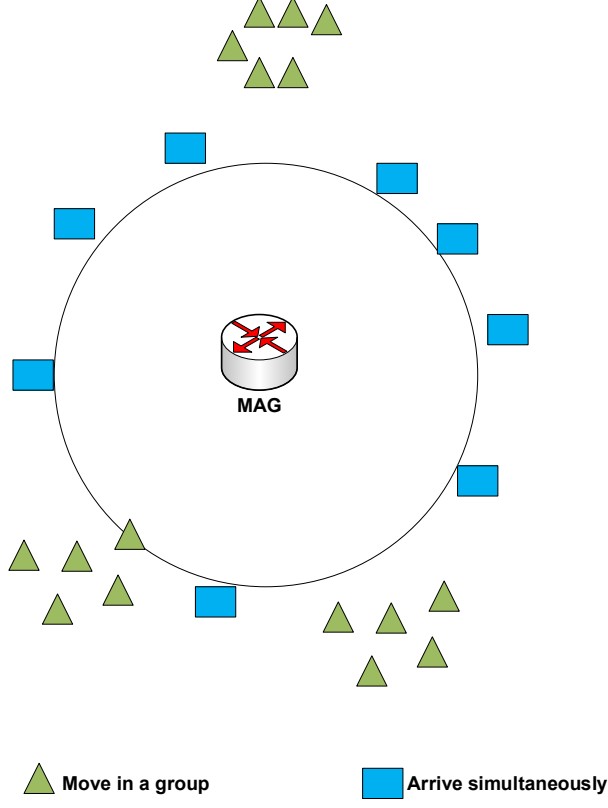

**Figure 1.** The MN' expectation scenarios.

To validate the proposed E-CPMIPv6 scheme, we compared the proposed E-CSPMIPv6 scheme with the recent Group-based Fast Handover (GB-FH) scheme [16]. Furthermore, the proposed scheme was also evaluated by comparison to the standard Proxy Mobile IPv6 (PMIPv6) protocol [7] and Cluster-based Proxy Mobile IPv6 (CSPMIPv6) protocol [12] mobility management protocols using numerical analysis and simulation experiments to show the superiority of the proposed E-CPMIPv6 scheme in terms of signalling cost, handover latency and end-to-end delay.

The contributions of this study are summarised as follows:

1.  A novel efficient clustering mechanism is introduced for grouping MNs moving simultaneously before processing their handoffs.
2.  A new mechanism is proposed for simultaneously manipulating the mobility-related signalling for a group of MNs that are triggering their handoff at the same time.
3.  A numerical analysis was performed to test the performance of E-CPMIPv6 in terms of the handoff latency, the analysis was validated by extensive simulations.

The remainder of this article is organised as follows: Section 2 presents an extensive overview of mobility management protocols highlighting their advantages and disadvantages. In Section 3, we propose the system model that is used as the environment for the E-CPMIPv6 scheme. Section 4 presents in detail the E-CPMIPv6 Scheme. In Section 5, we explain in detail the proposed E-CSPMIPv6 methodology within the localised mobility protocol environment. Section 6 discusses the performance evaluation of the E-CPMIPv6 Scheme. Finally, the study contributions are summarised in Section 7.

## 2. Mobility-Related Study

Several protocols have been proposed to perform handover process for several MNs in a concurrent manner.

The protocol in [12] presents an enhanced architecture to PMIPv6 named, Cluster Sensor PMIPv6 (CSPMIPv6) architecture. This enhancement attempts to tackle the bottleneck issues in PMIPv6 protocol by dividing the proxy mobile domain into sub-local domains. Each sub-domain groups MAGs into clusters, with each cluster being managed and controlled by a cluster head. In the CSPMIPv6 architecture, the LMA and MAG functionalities are similar to LMA and MAG in PMIPv6 protocol. The key characteristic of the Head MAG (HMAG) is to relieve the LMA from any local mobility management. In addition, the HMAGs reduces the handoff latency and provides a route-optimised path in intra-communication mobility. Nevertheless, the mobility in this protocol is performed individually for each MN that enters the CSPMIPv6 domain, which leads to an increasing in the signalling cost and handoff latency. Accordingly, an efficient mobility management scheme has to consider this issue in its original design. Thus, this research originally is designed to make a group of MNs act as a single MN when they are changing their point of attachment from one network to another network in order to reduce the signalling cost and the time needed for the handoff.

To enhance the efficiency of mobility management, a group-based technique is utilised in [16]. This solution introduces a new scheme, named the Group-based Fast Handover (GB-FH) control scheme. To perform a grouping, this scheme selects one MN to act as a guide for several neighbouring nodes during the handoff process. The main idea of GB-FH is to re-register the MNs that are predicted to move together. The MN guide prepares itself for the handover before the actual handover processes are performed. According to their analytical analysis and simulation results obtained, the GB-FH scheme measurably reduces the time needed for handover initiation, messages signalling and handover latency compared to the Fast Mobile Internet Protocol Version 6 (FMIPv6). The GB-FH scheme signalling, including the process of its MN registration are illustrated in detail in [16]. Despite all achievements recorded, the GB-FH still has several limitations: the number of the predictable handover MNs is restricted due to only grouping the MNs that belong to the same network. Furthermore, The MN does not consider the de-registration messages that might occur before registering the MN in the new network, which might increase the handover latency time. In addition, the GB-FH scheme

was originally designed according to the principle of the host-based protocols that require the MN to participate in the mobility-related signalling, which surely leads to system performance degradation.

The Correlated MNs Detection Algorithm (CMDA) is used to group MNs based on their Signal-to-Noise Ratio (SNR) and the history of MNs handoff information according to the network-based printable [17]. This algorithm is developed to reduce the handover and the signalling cost. The idea beyond this work is to group the MNs that approximately reside in the same physical area with the same SNR. The MAG always scans and sends the MN SNR periodically to the LMA in order to group them according to their SNR. Thus, when a MN from the grouped MNs starts its mobility signalling, the MAG sends a Proxy Binding Update (PBU) message to the LMA. The LMA then looks for that MN in its associated group of MNs. If the respective MN has been found, a Proxy Binding Acknowledgment (PBA) message is then sent to the MAG including the Home Network Prefix (HNP) for all the group members in order to accelerate their registration. Subsequently, when a MN movements belongs to this group is detected by a MAG, the MN exchanges a Router Solicitation (RS) message with the MAG. The MAG then directly delivers the HNP created earlier to the MN. Despite the landmark achievement with respect to the signalling cost and the handover latency, this work might group the MNs from different networks but with the same SNR, which increase the false MNs handoff predictions.

The work proposed in [18] introduces a fast handover through the group-based mechanism. The main goal of this scheme is to group the control messages of sensors that are distributed in the human body. The coordinator is responsible for carrying out the mobility for all the sensors by including sensor IDs within its RS message that is sent to the MAG. The MAG, upon receiving the RS message, starts authenticating each MN with an Authentication, Authorisation, and Accounting (AAA) server. When the authentication is performed successfully, the MAG sends one PBU with the sensor-IDs to the LMA to register the new connected sensors. Once this registration process is completed by the LMA by locating HNPs, a PBA reply message is then sent to the requesting MAG. From the simulation results obtained, this scheme greatly reduces the signalling cost and handover latency. However, This work does not consider the wireless area network, thus heavy signalling is still experienced in this work due to the periodic scanning performed by individual sensors to scan their activity. This work is dedicated to the sensors that reside in one physical area, which always move together.

A new group-based scheme, named a bulk Fast Handover for Proxy Mobile IPv6 (bFP-NEMO), is proposed in [19] to mitigate the signalling cost in the vehicular networks. This scheme is introduced to establish a tunnel for a group of vehicles instead of each single vehicle. The idea of this work is grouping the MRs when their link layer is detected by the MAG. Then, the MAG delivers an HI message to all the neighbouring MAGs listed in the neighbouring list to establish a pre-tunnel with these MAGs. The results from this simulation demonstrate an enhanced system performance which outperformed the counterpart protocols. However, this work is dedicated to the networks that are equipped with MRs. Furthermore, performing a tunnelling with all neighbours leads to an increase in the tunnelling overhead, especially in the false prediction situation that might occur when the MRs return back to its network. In addition, this work is limited by NEMO that may effect on supporting an efficient mobility for the MNs [15].

Another PMIPv6-based group mobility management protocol for IoT device is proposed in [15]. In this protocol, the authors enhanced the operation of PMIPv6 by introducing a multi-node handover method for IoT devices over PMIPv6. This is achieved by using the bulk binding update standard that defines multiple connections' handoff and revocation operations for a group of mobility sessions [14]. This solution groups the binding messages using some metrics such as movement similarity [13] in order to perform a binding for a group of IoT devices. As a result, the performance in terms of scalability and bandwidth is enhanced.

Another work, named Constrained Application Protocol based Group Mobility Management Protocol (CoAP-G), is proposed in [20] to support mobility management in a web-based Internet-of-Things environment. In this scheme, one sensor is responsible for transmitting the control messages of the body sensors to the web-of-things mobility management system (WMMS). The sensors' information is maintained by the WMMS. Two tiers of IP addresses are utilised: one for the sensor IP, which is the permanent address, and the other one for the Access Router (AR) address, which is a temporary address. According to the numerical analysis obtained, the CoAP-G has better system performance compared to CoAP protocol. This is due to the incorporation of sensors within the CoAP protocol, which decrease the cost of signalling and the handoff delay [21].

Xiaonan et al. [22] introduced vehicular networks based on vehicle trees for supporting mobility management. This is achieved by minimising the handover latency through the reduction of the CoA configuration latency and channel scanning latency. The authors utilised the Vehicle Tree (VT) to represent several vehicles in order to perform the handovers as a group. Thus, the total number of handovers is reduced as a result of using only one address configuration operation and only one channel scanning operation is performed for all the vehicles in a VT. From the results obtained, the performance of this scheme, which is subject to a low handoff latency, has been greatly reduced.

However, none of the aforementioned solutions efficiently address the case when a group of MNs belonging to the same network frequently performs the handover process intermittently. Thus, ignoring this scenario defiantly leads to several serious issues such as long handover, heavy signalling and high bandwidth cost. To address this problem, we propose a scheme that groups the MNs that roam together as well as the MNs that trigger their handover process simultaneously, thereby utilising an efficient control message scheme that exploits the above fact.

## 3. The Proposed E-CSPMIPv6 Scheme

This section presents the proposed E-CSPMIPv6 scheme highlighting its main advantages over the CSPMIPv6 and GB-FH schemes. The proposed E-CSPMIPv6 scheme introduces an efficient scheme for grouping the MNs' binding messages with a special consideration for minimising handover latency and signalling cost. In the proposed scheme, two mechanisms, named Clustered neighbouring MNs (CN-MN) and the Clustered Remote MNs (CR-MN) are introduced to efficiently enhance the mobility management process in the IP-WSNs. In the CN-MN mechanism, a clustering technique is used to group the neighbouring MNs which move simultaneously as a group between two different networks within the CSPMIPv6 domain, as shown in Algorithm 1. In this mechanism, every MN has to continually calculate the RSS value ($S_{min}$) and compare it with the pre-configuration threshold ($S_{th}$), as shown in Algorithm 1. If the MN's RSS value exceeds the threshold, this MN becomes a Head Cluster (HMN) with an ability to group the neighbouring MNs using the Request Joining (Req-join) and Accept Joining (Acc-join) messages. This is performed to register/de-register them in advance. The HMN, after becoming a head cluster, sends a broadcast message to its neighbouring MNs to form a cluster. Finally, the HMN classifies the successfully joined MNs based on their serving MAG into lists to send their requests to the related MAG to process connections/de-connections a priori.

---

**Algorithm 1:** HMN functionalities.

1 **Initialisation:**
2 Read $S_{min}$ threshold;
3 Read $S_{th}$ threshold;
4 Set list1 to Null "contains the MNs addresses of HMN's SMAG where SMAG is the Service MAG";
5 Set list2 to Null "contains the MNs addresses of HMN's TMAG where TMAG is the Target MAG";
6 Calculate the timing threshold using Equation (11);
7 **Event** *On Packet Reception* **do**
8 　Calculate the *RSS*;
9 　**if** *RSS* $>= S_{min}$ **then**
10 　　Change MN to HMN ;
11 　　Sends a broadcast Req-join message to all the MN neighbors;
12 　**end**
13 　**if** *HMN recieves Acc-join message* **then**
14 　　Checks the SMAG of the joined MN;
15 　　**if** *The joined MN's SMAG belong to the HMN's SMAG* **then**
16 　　　Add MN address to the list1;
17 　　**end**
18 　　**if** *The joined MN's SMAG belong to the HMN's TMAG* **then**
19 　　　Add the MN address to the list2
20 　　**end**
21 　**end**
22 　**if** *RSS* $= S_{th}$ **then**
23 　　Send RS message to both serving MAG and new MAG including list1 and list2 "to pre- and de-register them a priori";
24 　**end**
25 　**if** *RSS* $> S_{th}$ **then**
26 　　Stop grouping the MNs;
27 　**end**
28 **end**

---

At the MAG side, when the RS is received by the related MAG, the MAG updates its Binding Update List (BUL) and sends a PBU message to the related HMAG, which in turn sends this request to the associated LMA after performing the on side processing, as shown in Algorithm 2. The LMA updates its Binding Cash Entry (BCE). It further creates new prefixes for the HMN and its neighbouring MNs upon successfully receiving the LPBU message sent by the HMAG. Finally, the LMA forwards the created prefixes by sending an LPBA message to the related HMAG. The HMAG now sends it to the related MAG to deliver these new addresses to the connected MNs when they announce their presence. This mechanism is meant to overcome the issues associated with the GB-FH scheme mentioned earlier.

---

**Algorithm 2:** MAG, HMAG and LMA functionalities based on the CN-MN mechanism.

---

**1** **Event** *On RS received by the MAG* **do**

**2**     Updates its BUL table;

**3**     Sends an LPBU message to the corresponding HMAG to pre-/de-register the MNs;

**4**     **if** *HMAG receives the LBPU message* **then**

**5**         Updates its BUL table;

**6**         Sends an PBU message to the LMA to pre-/de-register the MNs;

**7**     **end**

**8**     **if** *LMA receives the PBU message* **then**

**9**         Updates its BCE;

**10**         Send an PBA message to the corresponding HMAG including the new HNPs;

**11**     **end**

**12**     **if** *HMAG receives the PBA message* **then**

**13**         Sends an LPBA message to the related MAG including the HNPs;

**14**     **end**

**15**     **if** *MAG receives the LPBA message* **then**

**16**         Sends a broadcast RA message to all the joining MNs "this is to inform the MNs about their successful joint and to send the HNP to the HMN";

**17**     **end**

**18**     **if** *HMN receives the RA message* **then**

**19**         Starts configure its new address " this means the MN has the ability to send the packets" ;

**20**     **end**

**21**     **if** *The neighbouring MN detected by the new MAG* **then**

**22**         Sends RS message to the new MAG "this done to inform its presence";

**23**     **end**

**24**     **if** *MAG receives RS of the neighbouring MN* **then**

**25**         Sends RA message to the neighbouring MN including its new HNP;

**26**         Sends LPBU to the HMAG including the presence MN address;

**27**     **end**

**28**     **if** *HMAG receives LPBU message and the MN belong to its domain* **then**

**29**         Activates the HNP and sends the packets to the new CoA;

**30**     **else**

**31**         Sends PBU to the LMA to activate the HNP of the presence MN;

**32**     **end**

**33** **end**

---

With the CR-MN mechanism, the MNs that arrive simultaneously to the same MAG are processed together to reduce the signalling cost, as shown in Algorithm 3. In this mechanism, the MAG simultaneously processes the mobility-related signalling of the MNs that request a new link at the same time. This is done by using one PBU message for all the connected MNs and sending this message to the related HMAG. All subsequent messages and their processing between the HMAG-LMA and LMA-HMAG are similar to the CN-MN mechanism. After the MAG receives the PBA sent by the HMAG, it sends the created HNPs to the new connected MNs using broadcast message (e.g., Media Access Control Address (MAC)) or individual RA message for each connected MN. E-CSPMIPv6 considers the CSPMIPv6's entities (e.g., LMA, HMAG, or MAG) without any modification to both mechanisms.

---

**Algorithm 3:** MAG, HMAG and LMA functionalities based on the CR-MN mechanism.

---

1 **Event** *On RSs receiving by the MAG* **do**

2 | Updates its BUL table;

3 | Sends an LPBU message to the corresponding HMAG to register the MNs;

4 | **if** *HMAG receives the LBPU message* **then**

5 | | Updates its BUL table;

6 | | Sends an PBU message to the LMA to register the MNs;

7 | **end**

8 | **if** *LMA receives the PBU message* **then**

9 | | Updates its BCE;

10 | | Send an PBA message to the corresponding HMAG including the new HNPs;

11 | **end**

12 | **if** *HMAG receives the PBA message* **then**

13 | | Sends an LPBA message to the related MAG including the HNPs;

14 | **end**

15 | **if** *MAG receives the LPBA message* **then**

16 | | Sends a broadcast RA message to all the handovers MNs "this is to send the HNPs to the MNs";

17 | **end**

18 | **if** *MN receives the RA message* **then**

19 | | Starts configure its new address " this means the MN has the ability to send the packets" ;

20 | **end**

21 **end**

---

*The flow diagram of the E-CSPMIPv6 scheme*

Figures 2 and 3 show the message flow diagram for the proposed E-CSPMIPv6 scheme according to the functions of the CN-MN and CR-MN mechanisms, respectively.

In the proposed E-CSPMIPv6 scheme, the MNs have to exchange some messages before the handover takes place in order to reduce the handover latency. This pre-exchange of messages is initiated to handle the control signalling of a group of MNs. This consequently shortens the handover latency and signalling cost.

The introduction of MNs in this study is very pertinent, as it improves the overall system performance in terms of handover latency and signalling cost. It thus makes the system suitable and deployable for critical real-time applications. Several studies (e.g., [23–25]) added extra functions (e.g., IEEE 802.21 Media Independent Handover (MIH)) to improve the system efficiency [26].

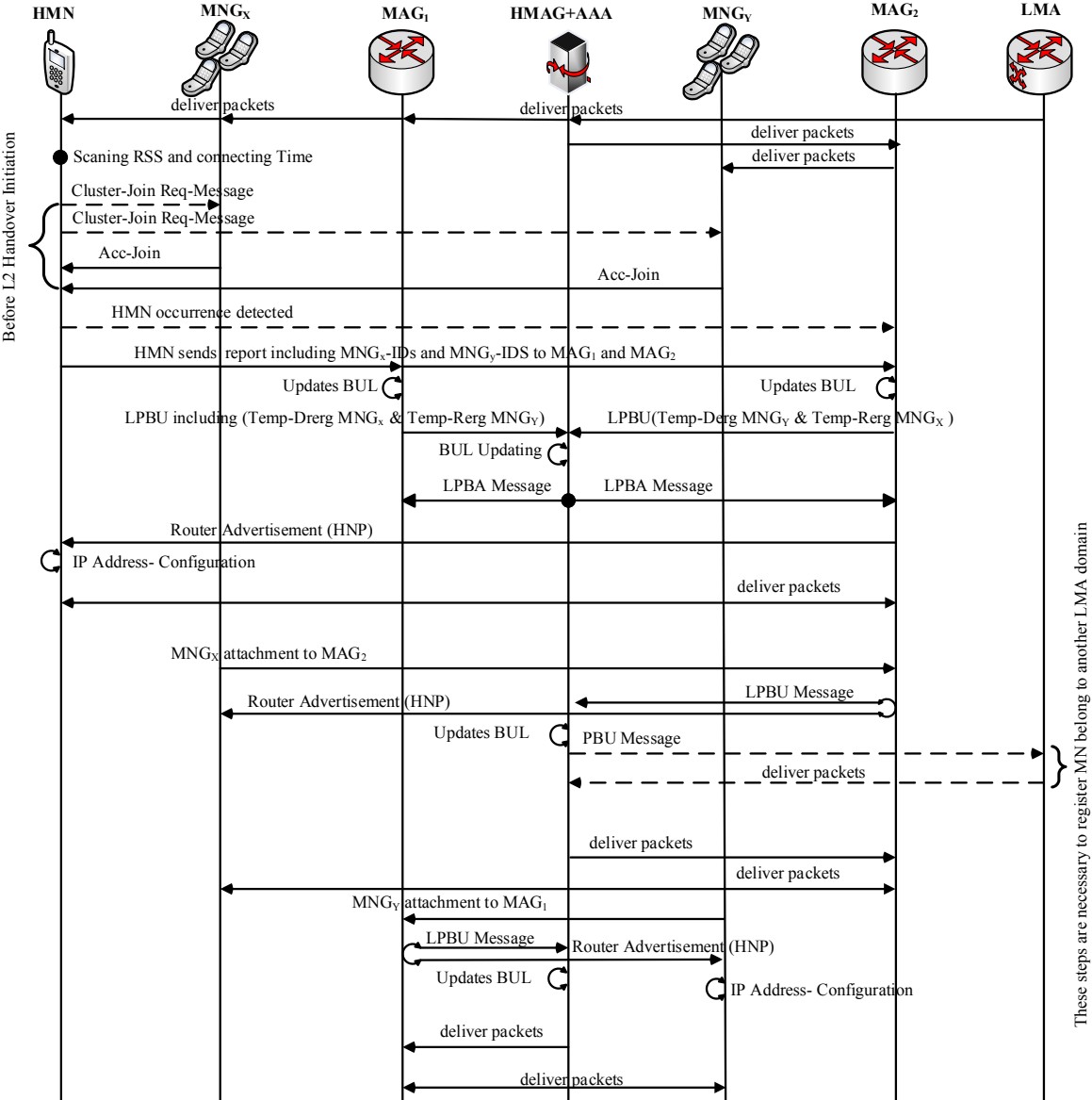

**Figure 2.** Message flow diagram for E-CSPMIPv6 scheme based on CN-MN mechanism.

In Figure 2, each MN continuously monitor its RRS for any change. When the RSS value surpasses the $S_{min}$ threshold, the MN sends Req-join to neighbouring MNs requesting them to form a cluster in which it acts as the cluster Head (HMN). The neighbouring MNs that join the cluster successfully become members of the cluster. To enhance the system performance in terms of handover latency and signalling cost, the proposed scheme reduces the cluster size (i.e., clustering the nearest neighbouring nodes) by minimising the Req-join coverage area. Moreover, the mechanism for grouping the MNs is performed at an appropriate time to reduce false prediction. This is achieved by setting $S_{min}$ threshold to measure the MNs' RSS from their old MAG and timing threshold value to measure their connected period time in order to prevent the recently connected MNs from being grouped, as depicted in Algorithm 1. This leads to minimal handover latency and signalling cost. After that, the MNs associated with same serving MAG, together with those associated with the target MAG respond by sending an Acc-join message to their HMN. Upon receiving the Acc-join messages, the HMN stores the senders' addresses (e.g., MAC addresses) in its database for future use (pre-registration/de-registration). Since the HMN receives Acc-join messages from MNs belonging to different networks, it keeps them in different lists based on their network. All the aforementioned messages are exchanged before the

handover takes place. Thus, there is no preparatory stage for the handover process (handover initiation and handover execution).

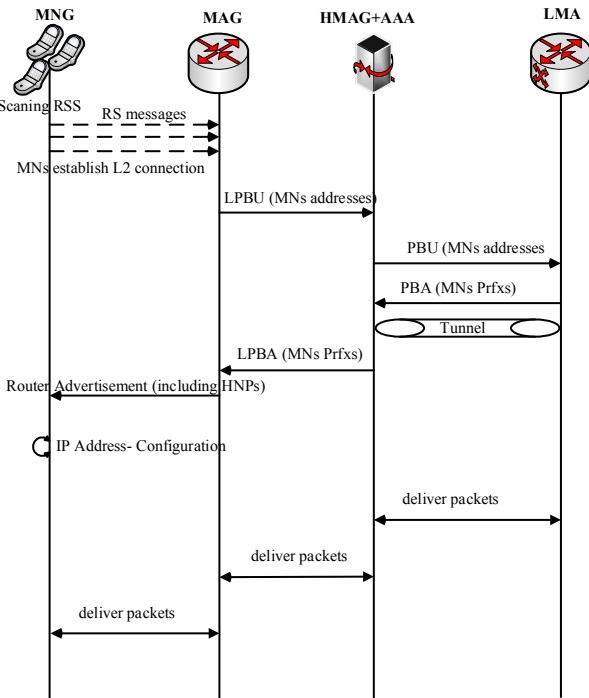

**Figure 3.** The message flow diagram for E-CSPMIPv6 scheme based on CR-MN mechanism.

When the HMN reaches $S_{th}$ threshold value, the HMN sends an RS message to the new MAG requesting to be registered. The RS message also carries information about the created list of the cluster members into its database, which is to be included in the registration process. The RS message is modified to carry the addresses stored in the HMN database lists. Moreover, the HMN sends a report message to the serving MAG and its new MAG, as in [27], to inform them to pre-register/de-register their members. In addition, the HMN keeps receiving new Req-join messages from new neighbouring MNs as long as these messages are coming from MNs that are still connected to their old MAG. This is performed in order to increase the opportunity to pre-register/de-register MNs, especially when the MNs come in a sequential order (sparse network).

Thereafter, the new MAG registers the HMN and temporarily registers the neighbouring MNs that belong to the serving MAG. It also temporary de-registers the neighbouring MNs that are near to leave its coverage area. This is done by sending a modified Local Proxy Binding Update (LPBU) message that is compatible with the proposed E-CSPMIPv6 scheme to the related Head MAG (HMAG). Similarly, the serving MAG temporarily registers/de-registers the MNs based on their CoAs. This temporary pre-register/de-register process increases the prediction accuracy.

After the HMAG receives the LPBU messages from serving MAG, the new MAG looks up the information of the HMN together with its associated members (i.e., its neighbouring MNs) and updates this information accordingly inside its database based on the request of the HMN and its neighbouring MNs.The HMAG sends a PBU message to its related LMA requesting that its HMN be registered, and the neighbouring MNs be temporarily registered/de-registered.

Once the LMA receives the request message, it updates its BCE for each MN based on the type of their registration. A new flag, named *S*, is used for the mobility-related signalling messages to distinguish between the temporary registration and the actual registration. In addition, the LMA sends a PBA message carrying the HNPs to the corresponding HMAG. Fields such as number of HNP options and the HNP options in the PBU and PBA messages are utilised to determine the requested number of HNP. Then, the corresponding HMAG sends a Local Proxy Binding Acknowledgment

(LPBA) message, which carries the HNPs of the MNs, to the requesting MAGs after updating its Binding Update table (BUL) table. When the MAGs successfully receive the LPBA message from the corresponding HMAG, the new MAG registers the HMN together with the neighbouring MNs that joined for temporary registrations/de-registrations. This registration is performed by the serving MAG and the new MAG updates its BUL tables according to the neighbouring CoAs. Subsequently, the new MAG sends a Router Advertisement (RA) message including the HMN's HNP and the addresses of the MNs that wish to join its network in a broadcast manner. The aim of the broadcasted RA message is to deliver the HNP to the HMN and to inform the neighbouring MNs about their successful joining. To group the MNs' signalling efficiently, the control messages are either extended such as RA and RS or the existing fields such as HNP are utilised. The original RS and RA control message that are used by the standard PMIPv6 protocol can be represented as Header, ICMP, MN-ID and Link-ID, where the Header contains the source and destination addresses, ICMP is a TCP/IP layer and MN-ID and Link-ID refer to MN-identifier and link-layer identifier, respectively. The Link-ID identifier contains a Header, ICMP and HNP, where HNP contains the MN's home network prefix. These messages are extended to carry several MNs' addresses and the new format become: Header, ICMP, MN-No, MN-ID1, Link-ID1, MN-ID2, Link-ID2, MN-IDn, Link-IDn, etc. The Reg-join and Acc-join messages are the same for RS and RA except that the link layer identifier represents the serving network address only. Similarly, there are multiple HNP options in the PBU message, and the number of HNP options indicates the amount of requested prefixes. The Prefix field of each HNP option is set to ZERO.

The HMN completes its registration processes by configuring its IP address when it receives the RA message from the new MAG successfully. Subsequently, any other neighbouring MN that has been temporarily pre-registered by the new MAG, sends an RS message to the new MAG to inform it of its presence. The new MAG sends an RA message to a specific neighbouring MN including the HNP. Similarly, it sends an LPBU message to the corresponding HMAG to activate the HNP of this MN. The HMAG activates the HNP of this neighbouring MN and then redirects the packets to the new CoA if the neighbouring MN belongs to this HMAG. Otherwise, the HMAG sends PBU to the LMA to activate the HNP of this neighbouring MN, as depicted in Algorithms 1 and 2. Therefore, the time required by the MN to configure its IP address is performed concurrently with the last two steps. Note that the time of the last two steps is negligible.

The proposed E-CSPMIPv6 scheme has several benefits with respect to its CN-MN mechanism, as illustrated below.

- The CN-MN greatly reduces the false prediction of MNs movement by clearly preventing newer MNs connected to the serving MAG from joining the cluster. This advantage can be justified by carefully observing Figure 4. As shown in Figure 4, the issue of the diamond interchange in the overlapping area that is covered by multiple MAGs is taken into consideration in the proposed E-CSPMIPv6 scheme. This is done by applying a time threshold value that prevents an MN from sending an Acc-join message if this MN has been connected to its serving MAG for a period less than the threshold value.
- The handover latency is reduced by eliminating the de-registration step from the handover process. Instead, the list created earlier by the HMN is sent to both MAGs (i.e., the serving and the new MAG) during the HMN handoff. The prior de-registration increases the system prediction accuracy by increasing the number of handoff MNs in the list prediction, which invariably reduces the handover latency and the signalling cost, and minimises bandwidth waste.
- The HMN keeps receiving the request joining messages after completing its registration processes until a predefined threshold is reached. This is applied to increase the pre-registration of the MNs as much as possible, especially in the sparse networks.

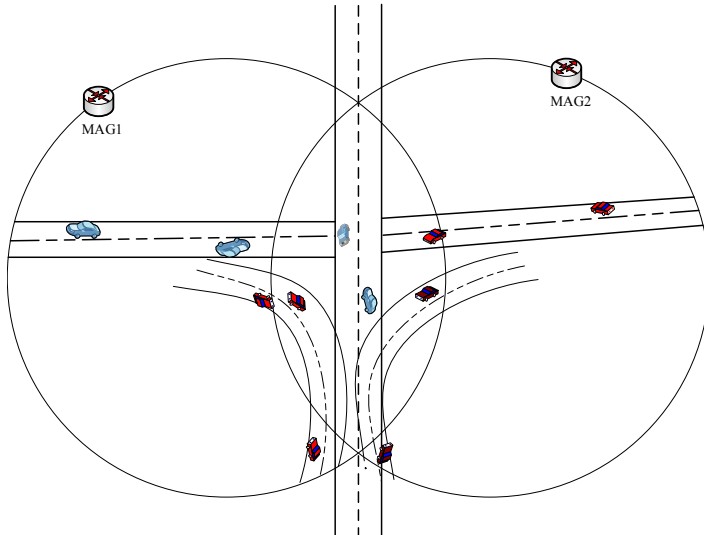

**Figure 4.** Example of diamond interchange road.

In the second mechanism of the proposed protocol (i.e., CR-MN), the MNs that arrive at the same time at the same MAG are considered, as shown in Figure 3. In the reviewed protocols in Section 2, the mobility-related signalling is performed for each MN, even if the MNs arrive simultaneously to the covering area of the same MAG. This increases both signalling cost and bandwidth waste. To address this issue, the proposed CR-MN mechanism allows simultaneous registration of several MNs that arrive at the same time, as illustrated next.

When the movements of several MNs are detected by the MAG at the same time, the MAG has two options for registering these MNs. In the first option, after the MAG detects the MNs movements, it immediately sends an LBPU message together with the MNs information to pre-register the detected MNs. In the second option, the MAG must wait for a while to collect the RS messages sent by the MNs that are detected by the MAG. The waiting time should be very short to avoid degradation in system performance. In the proposed mechanism, a prediction technique is used to alleviate the delay that could occur in the second option (i.e., timer). The scenario of this mechanism is implemented as follows:

In this mechanism, the $S_{th}$ threshold is employed by the MAGs to measure the RSS. When MNs are detected by the new MAG and the RSS $S_{th}$ threshold is reached, the MAG expects that the MNs will soon change their link layer. Subsequently, the MAG adds the addresses of these MNs to a list that has been created to group the MNs that are expected to perform handoff simultaneously. The MAG has to wait until one of these expected MNs, which is already added to the list, requires an actual handover. At this time, the MAG sends an LPBU including all the MAC addresses of the MNs within the list to the related HMAG to register them.

The HMAG, LMA and MAG behave similarly to CSPMIPv6 protocol with regards to registering the MNs by exchanging LPBU, PBU, PBA, LPBA and RA messages, as shown in Figure 3. Finally, the MNs receive the RA message, which is sent in a broadcast manner by the MAG to inform the MNs about their successful joining. When detected by the New MAG (NMAG), each MN informs the NMAG of its presence by the exchange of RS and RA messages. Accordingly, the NMAG sends the MN's HNP to the present MN to complete its registration (i.e., configures the new CoA).

In the CR-MN mechanism, it is clearly observed that the signalling cost and the burden on the bandwidth are reduced as a result of processing the mobility signalling for several MNs simultaneously.

## 4. System Models

Figure 5 shows the network model that was used for performance analysis, which represents a single LMA domain that encompasses an LMA, HMAGs, MAGs, and MNs. The HMAGs are attached

to LMA, MAGs are attached to its HMAGs, all via wired links. Only intra-domain communication and handoff operations are considered. Moreover, the abbreviations of parameters used for analysing the performance of the proposed scheme's model along with their definitions are described in detail in Table 1. In addition, all costs are symmetric, i.e., $TMN - MAG = TMAG - MN$.

**Table 1.** Parameters for the performance analysis.

| Parameter | Description |
|:---:|:---:|
| $T_{x-y}$ | Transmission cost of a packet between nodes x and y |
| $P_C$ | Processing cost of node C for binding update or lookup |
| $T_{setup}$ | Setup time for connecting MN with MAG |
| $N_G$ | Number of MAGs in PMIPv6 domain |
| $N_{HG}$ | Number of HMAGs in CSPMIPv6 domain |
| $N_{HM}$ | Number of active hosts per MAG |
| $N_{MH}$ | Number of MAGs per HMAG |
| $n$ | The probability number of MNs arrived simultaneously |
| $C_{x-y}$ | Hop count between nodes x and y |
| $S_{Ctrl}$ | Size of a control packet (byte) |
| $S_{Data}$ | Size of data packet (byte) |
| $a$ | Unit cost of binding update with LMA or HMAG |
| $b$ | Unit cost of lookup for MN at LMA, HMAG, or MAG |
| $t$ | Unit transmission cost of packet per a wired link (hop) |
| $k$ | Unit transmission cost of packet per a wireless link (hop) |
| $p$ | Probability of inter-cluster communications or movements |

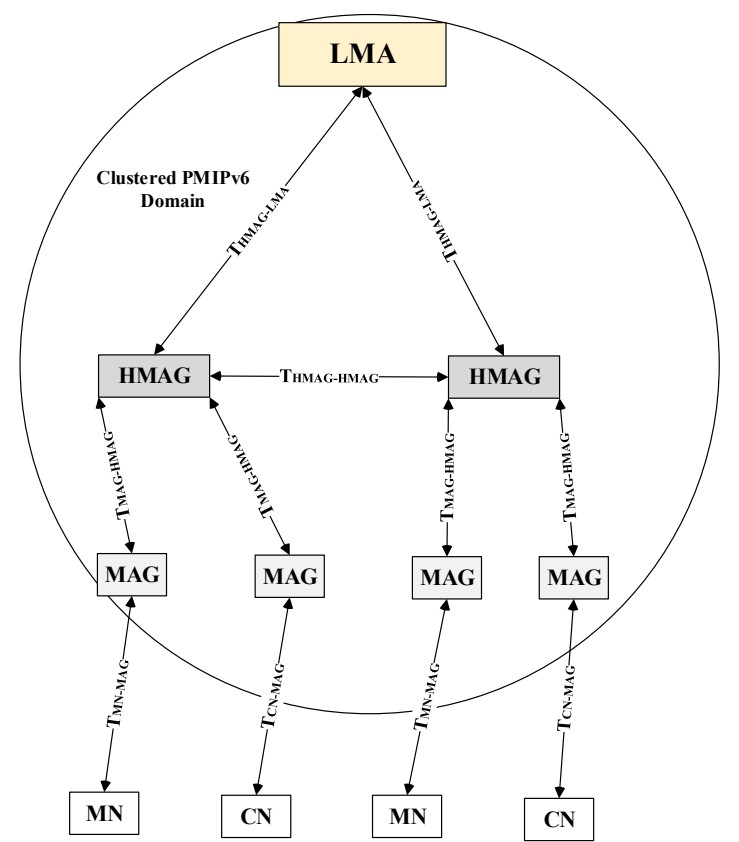

**Figure 5.** System Architecture of E-CSPMIPv6 scheme in localised domain.

Figure 6 shows the overlapping area inside the system model between the two MAGs region. The overlapped area was split into two main regions by applying threshold criteria. The specification and the notations are explained as follows:

- $R$ represents the circular radius that is covered by the MAG.
- $S_{th}$ represents the minimum threshold value of RSS at which an MN can consider joining and communicating with another MAG.
- $S_{min}$ represents the minimum threshold value of RSS, at which the MNs consider grouping the neighbouring MNs to apply pre-registration processes for the joined MNs.

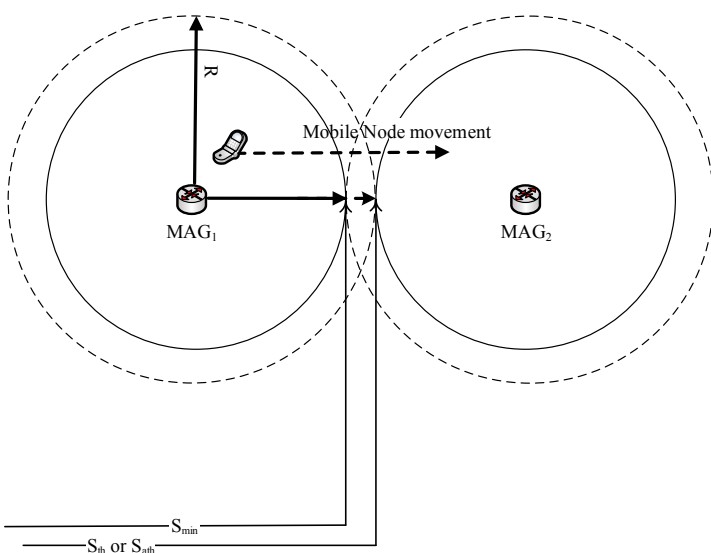

**Figure 6.** Overlapping area within the E-CSPMIPv6 infrastructure.

## 5. Numerical Analysis

In this section, we explain the performance evaluation of the proposed E-CSPMIPv6 solution and compare it with the existing PMIPv6 and CSPMIPv6 protocols and with the recent GB-FH scheme. Note that both numerical analysis and simulation were used in this study. We analysed the performance of the proposed scheme against the mobility management protocols in terms of handover latency, signalling cost and end-to-end delay.

### 5.1. PMIPv6 Cost Analysis

For purpose of analysis, the network model presented in [28] and modified in [12], was used with some modifications to calculate the Total Cost (*TC*). These modifications considered the number of the MNs that have been grouped in the mobility without the authentication cost. The *TC* is represented as:

$$TC = BUC + PDC, \tag{1}$$

where *BUC* represents the Binding Update Cost and *PDC* represents the Packet Delivery Cost.

Hence, several operations should be considered to perform a binding update process in the PMIPv6. The first operation is to setup the connection between the MN and its CoA (MAG), which takes $T_{setup}$. The second operation is authenticating the MN in both MAG and LMA, which requires $2T_{MAG-AAA} + 2T_{LMA-AAA}$, as explained in [12,29]. Finally, PBU and PBA messages must be exchanged between LMA and MAG to complete the registration processes of the MN, which take $2T_{MAG-LMA} + P_{LMA}$. Accordingly, the *BUC* of PMIPv6 can be calculated according to Equation (2).

$$
\begin{aligned}
BUC_{PMIPv6} = {} & T_{setup} + S_{Ctrl} \times (2T_{MAG-LMA} \\
& + 2T_{MAG-AAA} + 2T_{LMA-AAA}) + P_{LMA} = T_{setup} \\
& + S_{Ctrl} \times (2TC_{MAG-LMA} + 2TC_{MAG-AAA} \\
& + 2TC_{LMA-AAA}) + a\log(N_G \times N_{HM})
\end{aligned}
\tag{2}
$$

To deliver the packet created by the MN to its LMA, the packets should be forwarded to the related MAG and then to its LMA, which takes roughly $T_{MN-MAG} + T_{MAG-LMA}$. After that, the LMA fetches its policy database looking for the address of the CN, which requires $P_{LMA}$. Then, the LMA forwards the packets to the MAG serving the associated CN in order to send the packets directly to the CN. Accordingly, the *PDC* can be expressed as follows:

$$
\begin{aligned}
PDC_{PMIPv6} = {} & S_{Data}(T_{MN-MAG} + 2T_{MAG-LMA} \\
& + T_{MAG-CN}) + P_{LMA} = S_{Data}(kC_{MN-MAG} \\
& + 2tC_{MAG-LMA} + kC_{MAG-CN}) \\
& + b\log(N_G \times N_{HM})
\end{aligned}
\tag{3}
$$

### 5.2. CSPMIPv6 Cost Analysis

In the CSPMIPv6 protocol, the clustering technique is employed to divide the PMIPv6 domain into sub-local domains to support a seamless handover. Moreover, the authentication process is performed by the HMAG entity by utilising the LPBU and LPBA messages to reduce the signalling message cost. Two scenarios have emerged as a result of clustering in communications and movements. These scenarios are named intra/inter-cluster handoff and communication, respectively. This article concentrates on the movement scenarios to show the superiority of the proposed E-CSPMIPv6 scheme over CSPMIPv6 and PMIPv6 with respect to the handover latency, signalling cost and bandwidth usage.

#### 5.2.1. Intra-Cluster Handoff

In this scenario, the MN performs a handoff process between two MAGs that reside within the same cluster (i.e., having the same HMAG). Besides, the HMAG performs both the registration and authentication processes, which in turn doubles the processing cost ($2P_{HMAG}$). LPBU and LPBA messages are created and exchanged between the MAG and the HMAG to update the MN's location ($2T_{MAG-HMAG}$), since there is no need to access LMA in this scenario. Thus, the $BUC_{CSPMIPv6}^{Intra}$ can be expressed as:

$$
\begin{aligned}
BUC_{CSPMIPv6}^{Intra} = {} & T_{Setup} + S_{Ctrl} \times 2T_{MAG-HMAG} \\
& + 2P_{HMAG} = T_{Setup} + S_{Ctrl} \times 2tC_{MAG-HMAG} \\
& + 2a\log(N_{MH} \times N_{HM})
\end{aligned}
\tag{4}
$$

#### 5.2.2. Inter-Cluster Handoff

In this stage, the MN performs a handoff between two MAGs that belong to different clusters within the CSPMIPv6 domain. HMAG processing cost is doubled because HMAG performs both the authentication and registration functions ($2P_{HMAG}$). LMA involvement becomes necessary as it needs to update its BCE to achieve a successful handoff. The mobility-related signalling message is sent from the MAG to its HMAG, which in turn sends it to its LMA ($2T_{MAG-HMAG} + 2T_{HMAG-LMA}$). Thus, the BUC, in this case, can be expressed as:

$$
\begin{aligned}
BUC^{Inter}_{CSPMIPv6} = {} & T_{Setup} + S_{Ctrl} \times (2T_{MAG-HMAG} \\
& + 2T_{HMAG-LMA}) + 2P_{HMAG} + P_{LMA} = T_{Setup} \\
& + S_{Ctrl} \times (2tC_{MAG-HMAG} + 2tC_{HMAG-LMA}) \\
& + 2a\log(N_{MH} \times N_{HM}) + a\log(N_G \times N_{HM})
\end{aligned}
\tag{5}
$$

### 5.3. E-CSPMIPv6 Cost Analysis

In the CN-MN mechanism of E-CSPMIPv6 scheme, a clustering technique is employed to predict the handoff time for a group of MNs and to perform a mobility-related signalling prior to when MNs' change their point of attachment. This is performed by having the neighbouring MNs communicate with each other to form a cluster. Then, the cluster head pre-registers its member cluster nodes within the E-CSPMIPv6 domain. The binding update operations in the proposed E-CSPMIPv6 scheme are performed in a manner similar to those binding update operations that occur in the CSPMIPv6 protocol with some modifications. The binding update operations in PMIPv6 and CSPMIPv6 are performed for each MN willing to change its point of attachment. Note that the binding update operations are performed for several MNs simultaneously in the E-CSPMIPv6 scheme. Moreover, the mobility-related signalling messages (e.g., PBU, PBA, LPBU, LPBA, RA and RS) are modified to be compatible with our proposed scheme.

For the case of intra-cluster handoff, when prior successful registration information is received by a neighbouring MN, the neighbouring MN should inform the detected MAG about its presence by sending an RS message. Then, the new MAG sends the HNP to the present MN to set-up the connection between them ($T_{Setup}$). This is achieved by sending an RA message and an LPBU message to the HMAG at the same time to activate the MN's HNP. After that, the HMAG updates its BUL ($P_{HMAG}$) and begins to forward the packets to the MAG, which in turn forwards them to the present MN. In this case, there is no authentication process performed during the actual handoff by the HMAG because it has been performed in the pre-registration process. Accordingly, the BUC for intra-cluster handoff in CN-MN mechanism is calculated as follows:

$$
\begin{aligned}
BUC^{Intra}_{CSPMIPv6} = {} & T_{Setup} + S_{Ctrl} \times T_{MAG-HMAG} \\
& + P_{HMAG} = T_{Setup} + S_{Ctrl} \times tC_{HMAG-MAG} \\
& + a\log(N_{MH} \times N_{HM})
\end{aligned}
\tag{6}
$$

For the inter-cluster handoff, after receiving the LPBU message sent by MAG to the HMAG, the HMAG activates the located MN's HNP ($P_{HMAG}$) and sends a PBU message to the LMA ($T_{HMAG-LMA}$). The PBU is sent to activate the present MN's HNP in the LMA because the present MN comes from another cluster (another HMAG). Then, the LMA updates its BCE ($P_{LMA}$) and the packets are rerouted to the corresponding HMAG and then to the related MAG, which in turn forwards them to the present MN. Consequently, the BUC for inter-cluster handoff is measured as follows.

$$
\begin{aligned}
BUC^{Inter}_{CSPMIPv6} = {} & T_{Setup} + S_{Ctrl} \times (T_{MAG-HMAG} \\
& + T_{HMAG-LMA}) + P_{HMAG} + P_{LMA} = T_{Setup} \\
& + S_{Ctrl} \times (tC_{MAG-HMAG} + TC_{HMAG-LMA}) \\
& + a\log(N_{MH} \times N_{HM}) + a\log(N_G \times N_{HM})
\end{aligned}
\tag{7}
$$

where $P_{LMA}$ and $P_{HMAG}$ are the time of updating the MN's information inside the LMA's BCE and HMAG's BUL, respectively.

The communication cost between the MNs within CSPMIPv6 protocol is not affected by the proposed E-CSPMIPv6 scheme. Thus, the PDC is not influenced by applying the proposed E-CSPMIPv6 scheme within the CSPMIPv6 domain. Accordingly, the $PDC_{E-CSPMIPv6}$ for intra/inter-cluster communication is the same as the $PDC_{CSPMIPv6}$ intra/inter-cluster communication, as shown in Equation (8)

$$PDC_{CSPMIPv6} = PDC_{E-CSPMIPv6} \tag{8}$$

In the CR-MN mechanism, the E-CSPMIPv6 manipulates the mobility-related signalling of MNs to arrive together during both intra/inter-cluster handoffs. During intra-cluster handoff, when the MNs establish a link layer connection with the target MAG, RS messages are sent to the target MAG. The MNs also wait for the RA message in order to setup the connection ($T_{Setup}$). Upon receiving the RS messages, the target MAG sends one LPBU message to the corresponding HMAG containing the addresses of all detected MNs, which takes ($T_{MAG-HMAG}$). The corresponding HMAG responds to the target MAG upon successfully receiving the LPBU message by sending an LPBA message containing the HNPs for all the MNs ($T_{HMAG-MAG}$). Then, the target MAG sends RA message together with the HNPs of MNs in a broadcast manner to inform them about their HNPs in order to establish a new connection. Finally, when the MNs receive the RA message, they start performing IP address configuration to complete their connections. Accordingly, the BUC is measured as follows:

$$\begin{aligned}
BUC_{CSPMIPv6}^{Intra} &= T_{Setup} + S_{Ctrl} \times 2T_{MAG-HMAG} \\
&+ P_{HMAG} = T_{Setup} + \frac{S_{Ctrl} \times 2tC_{MAG-HMAG}}{n} \\
&+ a \log(N_{MH} \times N_{HM})
\end{aligned} \tag{9}$$

where $n$ is the predicted number of simultaneously arriving MNs.

In the inter-cluster handoff, the LMA involvement is necessary to complete the MNs registration. Thus, an extra message is exchanged between the HMAG and LMA in order to create a new HNPs for the MNs which takes ($2T_{HMAG-LMA} + P_{LMA}$). Consequently, the BUC is expressed as follows:

$$\begin{aligned}
BUC_{CSPMIPv6}^{Inter} &= T_{Setup} + S_{Ctrl} \times 2T_{MAG-HMAG} \\
&+ 2T_{HMAG-LMA} + \times P_{HMAG} + P_{LMA} = T_{Setup} \\
&+ \frac{S_{Ctrl} \times (2tC_{MAG-HMAG} + 2tC_{HMAG-LMA})}{n} \\
&+ a \log(N_{MH} \times N_{HM}) + a \log(N_G \times N_{HM})
\end{aligned} \tag{10}$$

## 6. Performance Evaluation

In this section, we detail the performance evaluation of the proposed solution E-CSPMIPv6 and compare it with the standard PMIPv6, CSPMIPv6 protocols and GB-FH scheme. This study was performed numerically and also by simulation. We analysed the performance of the proposed scheme against the mobility management protocols in terms of handover latency, signalling cost and end-to-end delay. First, we describe the simulation scenario and show the obtained results.

### 6.1. Numerical results

In this subsection, the numerical results are presented according to the explanation in Section 5. The numerical results were obtained based on the values depicted in Table 2. To accurately compare the proposed E-CSPMIPv6 scheme with CSPMIPv6 and PMIPv6 protocols, the same assumptions and parameter values used by Jabir et al. [12] were used in this study.

**Table 2.** Parameter values.

| Parameter | Description |
|---|---|
| $T_{setup}$ | 500 ms |
| $N_G$ | 20 |
| $N_{HG}$ | 4 |
| $N_{HM}$ | 200 |
| $N_{MH}$ | 5 |
| $n$ | 0 |
| $S_{Ctrl}$ | 50 byte |
| $S_{Data}$ | 1024 byte |
| $a$ | 3 |
| $b$ | 2 |
| $t$ | 2 |
| $k$ | 4 |
| $p$ | 0.5 |
| $C_{MAG-LMA}$ | 5 |
| $C_{HMAG-LMA}$ | 5 |
| $C_{MAG-HMAG}, C_{HMAG-HMAG}$ | $\sqrt{1 + N_{MH}}$ |
| $C_{MN-MAG}, C_{CN-MAG}$ | 1 |

Figure 7 shows the TC of E-CSPMIPv6 scheme, PMIPv6 and CSPMIPv6 protocols. To show the effect of the inter-cluster handoff on the TC, *P* parameter was varied between 0 and 1, while the other parameters were set to the default values. It can be observed in the figure that PMIPv6 has a constant performance in terms of TC regardless of the value of *P*. This can be attributed to the fact that the PMIPv6 performs the same operations regardless of the type of handover communication processes (i.e., intra/inter handover). On the other hand, the CSPMIPv6 shows a better performance compared to PMIPv6 protocol in terms of the TC. The amount of improvement is associated with the number of inter-cluster operations. Even when *P* is 1 (i.e., all the MNs perform inter-cluster handoff operations), the CSPMIPv6 still outperforms the PMIPv6 protocol. This can be attributed to the low handover latency and the optimal communication path. In contrast, the proposed E-CSPMIPv6 scheme shows a lower TC compared to CSPMIPv6 and PMIPv6 protocols. This is due to the elimination of some signalling operations during the handover process. This includes the authentication process and LPBA/PBA messages, which are performed in the pre-registration processes.

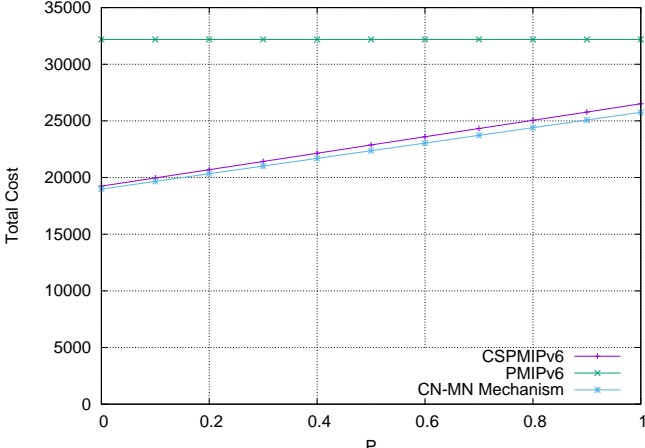

**Figure 7.** Effect the inter-cluster operation on the TC.

Figure 8 compares the two protocols with the proposed E-CSPMIPv6 scheme in terms of the TC by varying the number of MN arriving at the same time. The probability of the number of MNs arriving

at the same time was varied between 0 and 1 (where 0 implies no MNs are arriving simultaneously and 1 implies all MNs arrive at the same time). It was observed that the proposed scheme outperforms the other protocols. This superiority comes as a result of the group registration of all MNs arriving simultaneously. Moreover, the CSPMIPv6 shows a higher TC than the proposed E-CSPMIPv6 even as all MNs perform an intra-cluster handoff. This is because the CSPMIPv6 must register every MN separately. Finally, the PMIPv6 protocol shows the worst TC because each MN registered individually.

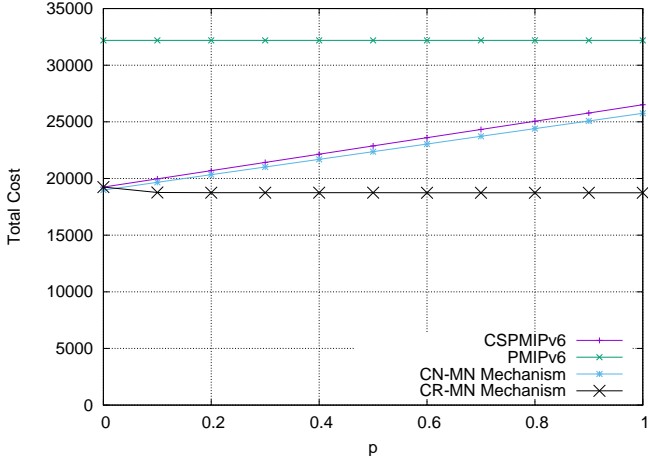

**Figure 8.** The overall TC of inter-cluster operation for CR-MN mechanism.

### 6.2. Simulation Scenario

The basic PMIPv6 architecture consists of a single LMA and a number of MAGs connected to it. To generalise the simulation model, a simulation scenario was introduced to study and analyse the performance metrics of this work. However, to simulate the E-CPMIPv6, the MAGs were divided into four clusters, each with six MAGs and one HMAG. All these HMAGs were connected to one LMA. MNs were distributed randomly in the simulation area along with a number of corresponding nodes outside the LMA domain, as shown in Figure 9. Random-walk mobility pattern was used in this work. In the mobility pattern, the MN keeps changing its velocity and direction during its motion within the LMA domain.

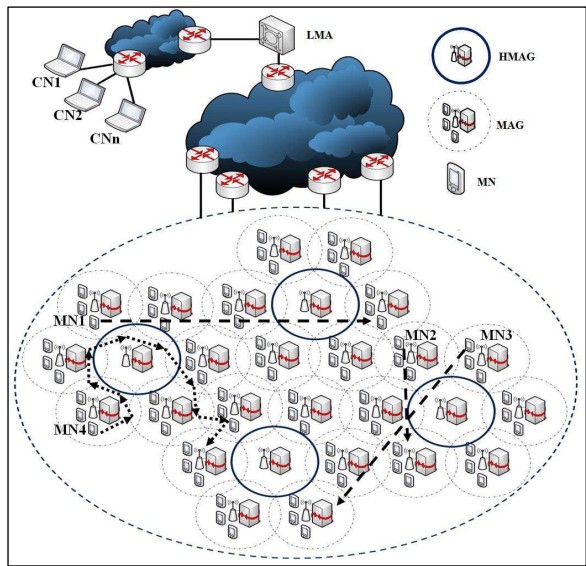

**Figure 9.** Simulation scenario [12].

### 6.3. System Setup

This subsection illustrates the simulation setup used in the experiments to evaluate the proposed load balancing mechanism. The proposed E-CSPMIPv6 was implemented on top of CSPMIPv6 environment in Network Simulator 2 (NS2) [30,31]. The test model consisted of one LMA, four HMAGs and six MAGs. The MNs were attached to the MAGs and the Corresponding Node (CN) was attached to the LMA. Wired links were used to connect the MAGs with HMAGs and the HMAGs with the LMA, where the link delay was set to 1 ms. The MNs used the wireless connection to enable them communicate with the MAGs. The MNs were distributed randomly among the MAGs and they moved according to the random-walk mobility pattern. In addition, the CN utilised the wired link for connecting itself with the LMA by a 2 ms link delay. The dimension of the topography was 3000 m $\times$ 2500 m, while the MAGs coverage area was set to 250 m. User Datagram Protocol (UDP) was used by the MNs and the CN to generate the packets with a size of 1024 bytes. The simulation time was set to 200 s and the interval between the successfully created packet and the next packet was 0.001 s. The number of MNs was varied between 10 and 100. These MNs moved between two roads in the opposite directions toward the MAGs with a speed of 20 m/s. The NS2 default parameter values of the drop-tail queue, two-ray-ground propagation model and the Destination-Sequenced Distance Vector (DSDV) Protocol were used for the wireless MNs. To consider roads similar to diamond-shaped roads, the coverage area of the MAG and the MN speed were used to calculate the timing threshold. In this scheme, the timing threshold was calculated using Equation (11). The constant number in the timing threshold equation is adjustable with respect to the movements of vehicles.

$$T_s = \left( \frac{C_{area}}{V_{Speed}} * \frac{1}{2} \right) \tag{11}$$

where $C_{area}$ and $V_{Speed}$ represent the MAG covering area and MN speed, respectively.

### 6.4. Simulation Results

To compare the proposed E-CSPMIPv6 performance with CSPMIPv6 and the basic PMIPv6 protocols, simulation experiments were performed and the results are shown in this subsection. The simulation parameters are presented in Table 3.

**Table 3.** Simulation parameters.

| Parameter | Description |
|---|---|
| Number of MNs | 10–100 |
| Network Area | 3000 $\times$ 2500 m |
| Simulation Time | 200 s |
| Node velocity | 1–50 m/s |
| number of MAG | 1–20 |
| number of HMAG | 1–4 |
| Packet Size | 1000 byte |
| Control packet Size | 68 byte |
| Agent | UDP |
| Traffic Type | CBR |
| Wired Link delay | 1–11 |
| Wired Link Bandwidth | 100 Mbps |
| Transmission Range | 500 m |

Figure 10 shows the signalling cost during the MNs handoff process in CN-MN mechanism using random distribution. The MNs number was varied within 10–100, while the wired link delay was set to 2 ms. It can be seen that the proposed E-CSPMIPv6 scheme has the lowest signalling cost

according to the principle of CN-MN mechanism. This is attributed to the fact that the proposed E-CSPMIPv6 scheme has better utilisation of the network channel as it encompasses the MNs' mobility-related signalling in one message unlike the CSPMIPv6 and PMIPv6 protocols. This is achieved because the CN-MN previously registers the MNs belonging to the target MAG and the serving MAG simultaneously. In addition, the CN-MN reduces the false prediction as a result of using the $S_{th}$ threshold. The GB-FH scheme fails to outperform the proposed scheme due to the high false prediction performed by the guide MN. Finally, the CSPMIPv6 and the standard PMIPv6 have the highest signalling cost. This is because both protocols do not support mobility for a group of MNs simultaneously.

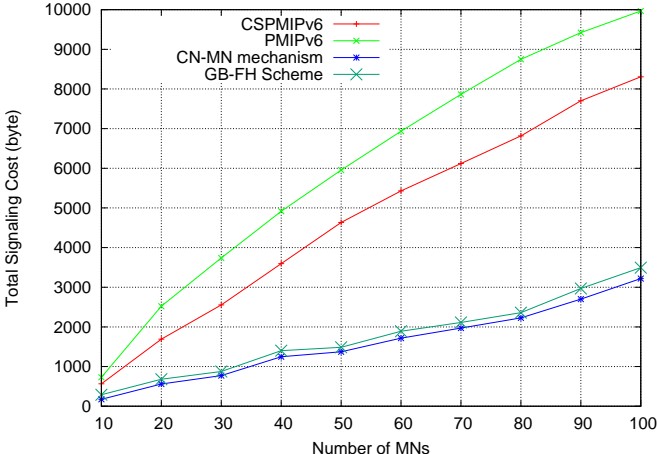

**Figure 10.** Performance of signalling cost vs. the number of MNs (CN-MN mechanism).

The superiority of E-CSPMIPv6, which employs the CR-MN mechanism, is observed in Figure 11. This is due to the grouping of the MNs that move separately. These results are different from the point-to-point case due to the random distributions of MNs among the MAGs, which may affect the number of MNs arriving simultaneously. On the other hand, the CSPMIPv6 performs better than the standard PMIPv6 because the CSPMIPv6 relieves the LMA from the intra-handoff mobility. However, the standard PMIPv6 and CSPMIPv6 protocols have the lowest performance due to the exchange of PBU and PBA messages for every MN that moves from one network to another.

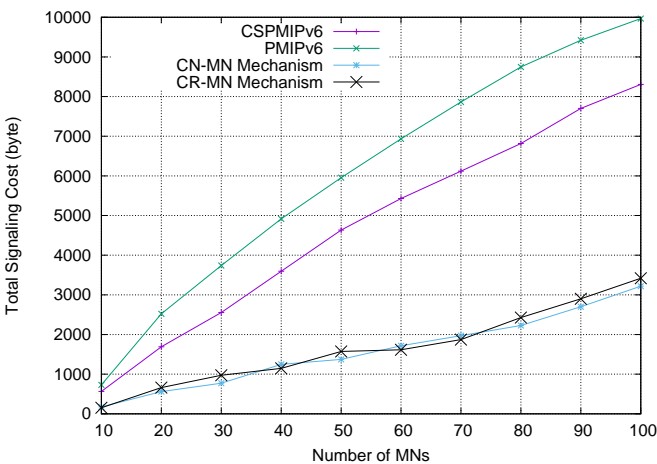

**Figure 11.** Performance of signalling cost vs. the number of MNs (CR-MN mechanism).

Figures 12 and 13 show the handover latency time for the proposed E-CSPMIPv6 scheme under CN-MN and CR-MN mechanisms, respectively, as compared to the CSPMIPv6 amd PMIPv6 protocols and the GB-FH scheme. The number of MNs was varied within 10–100, whereas the delay on wired link was set to 2 ms. It is obvious that the performance of the proposed E-CSPMIPv6 scheme in both mechanisms performs better than CSPMIPv6 and PMIPv6 protocols and the GB-FH scheme in terms of handover latency. The high performance of the proposed scheme can be attributed to the simultaneous pre-registration of several MNs, which reduces the handoff time. Moreover, the proposed E-CSPMIPv6 scheme, based on the CR-MN mechanism, achieves higher performance than the E-CSPMIPv6 scheme that is based on the CN-MN mechanism. This is due to the simultaneous processing of the higher number of MNs attached to the MAG coverage area. In contrast, the GB-FH performs better when compared with the CSPMIPv6 and PMIPv6 protocols. This is because the signalling messages of several MNs are grouped. However, the GB-FH fails to outperform the proposed scheme due to its false prediction that occurs as a result of accepting request messages from neighbouring MNs moving in the opposite direction. This false prediction leads to performing a handover for the MN individually. The PMIPv6 and CSPMIPv6 protocols register only one node at the time of their actual handover, which increases handover latency. In addition, the results show that the PMIPv6 has the lowest performance as it relies on the LMA to perform the handoff process. In addition, it employs the PMIPv6 to carry out a mobility signalling process for each MN that changes its network. This figure also shows that the CSMIPv6 outperforms the PMIPv6 because it carries out the intra-handoff process without involving the LMA. However, the inter-cluster handoff within the CSPMIPv6 protocol causes a significant delay due to the movement of MNs between different clusters. In fact, the movement of MNs between clusters requires the exchange of Proxy Binding Query (PBQ), and Proxy Query Acknowledgement (PQA) messages for carrying MNs information between the New HMAG (NHMAG) and the Old HMAG (OHMAG) which increases the total handover latency.

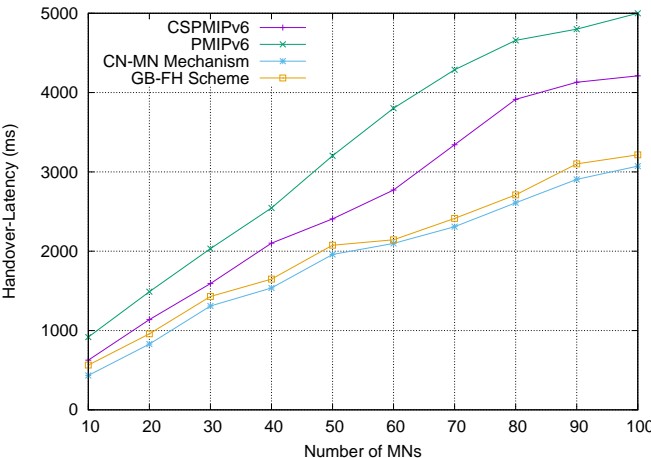

**Figure 12.** Performance of handoff latency vs. the number of MNs (CN-MN mechanism).

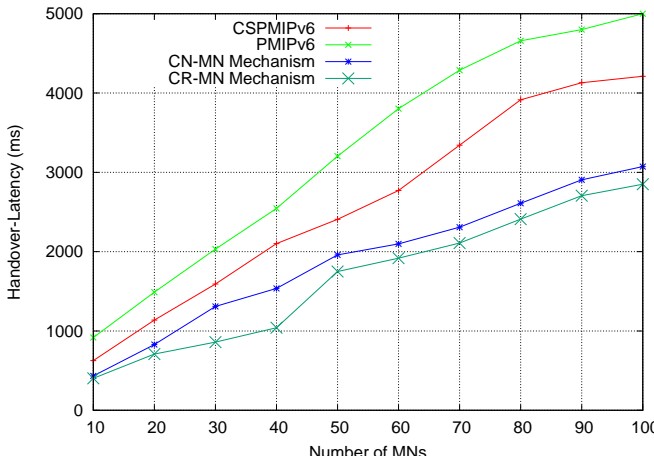

**Figure 13.** Performance of Handoff latency vs. the number of MNs (CR-MN mechanism).

Figure 14 depicts the end-to-end delay against the number of MNs for mechanisms 1 and 2, respectively. In both mechanisms, the delay increases when the number of MNs increases. The end-to-end delay of CN-MN mechanism has a lower performance than that of CR-MN mechanism. This is attributed to the fact that the packets suffer more contentions and thus more MAC layer retransmission. In other words, the number of times signalling is performed by the MNs during their handoff affects the queuing performance, which invariably increases the end-to-end delay. CR-MN mechanism considers the scenario where the MNs arrive simultaneously at the same target MAG regardless of their roads. This increases the number of MNs within the cluster, which in turn leads to a reduction in both signalling cost and handover latency.

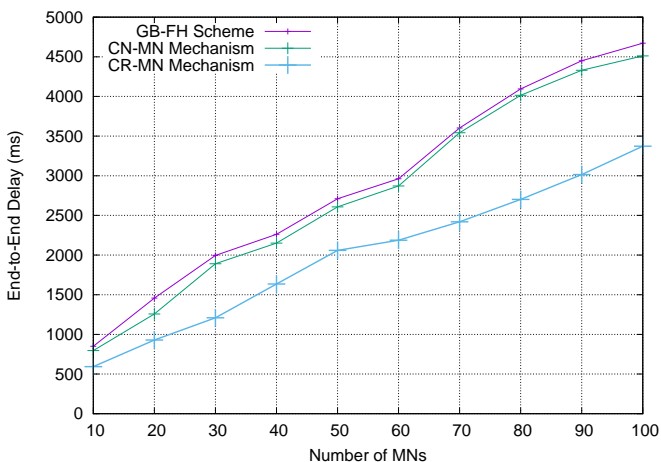

**Figure 14.** Performance of end-to-end delay vs. the number of MNs.

## 7. Conclusions

In this article, an efficient E-CSPMIPv6 scheme is introduced to provide a seamless handoff within the PMIPv6 domain. This is done by manipulating the handoff operations for several MNs simultaneously based on the CN-MN and CR-MN mechanisms.

In the CN-MN mechanism, pre-registration/de-registration processes are performed for the MNs that move in proximity with each other. The MN, which violates the pre-defined threshold requirements clustering with its neighbouring MNs, eventually becomes the cluster head. Thereafter, the cluster head takes the responsibility of performing the initial binding registration processes for its members. The mobility-related signalling messages such as RS and RA have been extended to carry

multiple addresses. The HNP and a number of HNP options fields in PBU, LPBU, PBA and LPBA messages are used. The grouping of MNs into clusters ensures low signalling, reduced end-to-end delay and low handover latency.

In the CR-MN mechanism, the mobility-related signalling messages are processed simultaneously for several MNs, which perform the L2 handover at the same time as the new MAG. This is done by grouping the MNs' L2 request messages by the new MAG and sending these requests as one message to the other CSPMIPv6 network entities. The bandwidth overhead is reduced by utilising the HNP and number of HNP option fields during the exchange of binding messages between the network entities.

Finally, numerical analysis and simulation results demonstrate that the proposed E-CSPMIPv6 scheme produces better performance in terms of handoff latency and signalling cost in comparison with the PMIPv6 standard protocol, CSPMIPv6 protocol and GB-FH scheme.

**Author Contributions:** Conceptualization, data curation, Formal analysis, investigation, methodology, software, validation, and writing—original draft preparation, S.M.G.; methodology, supervision, software, resources and Writing—review and editing, S.S.; software, validation, visualization and Writing—review and editing, M.G.; validation, Writing—review and editing A.M.E.E.

**Funding:** This research received no external funding.

**Conflicts of Interest:** The authors declare no conflict of interest.

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
