# Peer review of "An Efficient Group-Based Control Signalling within Proxy Mobile IPv6 Protocol"

_computers, doi:10.3390/computers8040075_

Round 1
Reviewer 1 Report
The authors in this work study a problem of integrated IPv6 with IoT/WSN. This is good idea. They propose a method based on the dynamically forming a cluster by the mobile nodes in some area. So, the paper needs more improvement in prsentation form and the related work.
So in prsentation form, some error must be corrected like 2. mobility-related study --> 2. Mobility-related study, ....
The section References should be added.
RSS must be defined in the introduction istead in the algorithm.
......
Author Response
Dear Reviewer,
I am writing, on behalf of the authors, to inform you that we are resubmitting an article for your consideration to be published in Computers journal. The title of the article is ” An Efficient Group-based Control signalling within Proxy Mobile IPv6 Protocol”. The article is the result of our research undertaken in extensive simulations within the network-based protocols.
Based on your previous comments, we have revised and rewritten the article considering your proposed comments. The list of changes is presented below.
I hope the listed corrections will rise up to your satisfaction. Thank you in advance.
Yours sincerely,
Safwan Ghaleb Corresponding Author
Reviewer #1
(1) The authors in this work study a problem of integrated IPv6 with IoT/WSN. This is good idea. They propose a method based on the dynamically forming a cluster by the mobile nodes in some area. So, the paper needs more improvement in prsentation form and the related work.
So in prsentation form, some error must be corrected like 2. mobility-related study --> 2. Mobility-related study, ….?
A (1) Thank you for this valuable comment. We have carefully revised the manuscript and we have taken special care to the writing language, Sections, figures and tables
(2) The section References should be added.?
A (2) thank you for this valuable point. The references Section is added.
(3) RSS must be defined in the introduction instead in the algorithm.?
A (3) Thank you for this great point. The RSS definition is already added to the section of the introduction, Actually, we use two terms RSS and SNR, which can be used interchangeably in this paper and both they are identified in the introductory part.

Reviewer 2 Report
In this paper, authors propose an efficient group-based handoff scheme, named Enhanced Cluster Sensor Proxy Mobile IPv6 (E-CSPMIPv6), in order to reduce the nodes handover during their roaming. This scheme E-CSPMIPv6 implement CN-MN and CR-MN mechanisms. The efficiency of the proposed scheme is validated through extensive simulation experiments and numerical analyses. The results illustrate that the E-CSPMIPv6 significantly improves the overall system performance, by reducing handover delay, signalling cost and end-to-end delay.
Authors have done an interesting work in this paper. Reviewer believes that their work can be a good contribution for Group-based mobility management within PMIPv6. However, reviewer has some major concerns about the current manuscript, and has suggestions, which might help them to improve their current draft.
In section 3, please clearly describe the proposed scheme. The text description is not consistent with fig.2, fig.3, fig.4, algo.1 and algo.2. The format of Reg-join and Acc-join should be presented. RS, RA, HNP should be explained.
In section 5, the Parameter P (MNs perform inter-cluster handoff operations) is not applied.
In section 6.1, please compare the performance with the varied number of MNs.
In section 6.2, please compare the performance with different parameter P.
Page 2, line 56, please give a reference and compare Cluster-based Proxy Mobile IPv6 (CSPMIPv6) with E-CSPMIPv6 in section 2.
Author Response
Manuscript Number: computers-537173
Manuscript Title : An Efficient Group-based Control signalling within Proxy Mobile IPv6 Protocol
Dear Reviewer,
I am writing, on behalf of the authors, to inform you that we are resubmitting an article for your consideration to be published in Computers journal. The title of the article is ” An Efficient Group-based Control signalling within Proxy Mobile IPv6 Protocol”. The article is the result of our research undertaken in extensive simulations within the network-based protocols.
Based on your previous comments, we have revised and rewritten the article considering your proposed comments. The list of changes is presented below.
I hope the listed corrections will rise up to your satisfaction. Thank you in advance.
Yours sincerely,
Safwan Ghaleb Corresponding Author
Reviewer #2
1. In section 3, please clearly describe the proposed scheme. The text description is not consistent with fig.2, fig.3, fig.4, algo.1 and algo.2. The format of Reg-join and Acc-join should be presented. RS, RA, HNP should be explained.?).
A (1) Thank you for highlighting this great point. We have carefully revised the manuscript and we have taken special care to the writing language, Sections, figures, and table. In addition, more explanation in Section 3 was added with respect to the control messages (Reg-join and Acc-join , RS, and RA) in order to clearly understand the formatting of these messages.
2. In section 5, the Parameter P (MNs perform inter-cluster handoff operations) is not applied.?
A (2) We thank you for bringing up this point. Yes, you are right. Only the intra-cluster is considered in this study as we mentioned in Section 4 line 341 “Only intra-domain communication and handoff operations are considered”. This is because the intra-cluster represent the optimal case of the benchmarks. Hopefully, in the next work, we will consider this great point.
3. Some acronyms need to be listed in full when used first, e.g. is MN short for moving node in the abstract?AE-PMIPv6, what is AE short for?
A (3) Thank you for this great point. We have corrected the acronyms in the entire text. AE-PMIPv6 is an Enhanced PMIPv6 scheme. Adding A is used to provide the distinction of difference for further clarity between the benchmark E-PMIPv6 and the enhanced proposed scheme.
4. In section 6.1, please compare the performance with the varied number of MNs.?
A (4) We thank the reviewer for the constructive feedback. The number of MNs that was used is 200 . These MNs are varied automatically according to the P value. Example, If the P value 90%, that means 90% of MNs perform an inter-cluster handoff while the others perform an intra-cluster handoff.
5. In section 6.2, please compare the performance with different parameter P.?
A (5) We thank the reviewer for the valuable feedback. P is a probability value, which represents the value between 0 and 1 (where 0 implies no MNs are arriving simultaneously and 1 implies all MNs arrive at the same time). The number of P value is changed depending on the number of MNs that arrive simultaneously or based on the number of the MNs that performing an inter- cluster handoff, as explained in the results.

Round 2
Reviewer 2 Report
The proposed E-CSPMIPv6 implement CN-MN and CR-MN mechanisms. Through extensive simulation experiments and numerical analyses, the efficiency of the proposed scheme is validated. The results illustrate that the E-CSPMIPv6 significantly improves the overall system performance, by reducing handover delay, signalling cost and end-to-end delay.